# OpenReview forum: "Is Vibe Coding Safe? Benchmarking Vulnerability of Agent-Generated Code in Real-World Tasks"
_ICML.cc/2026/Conference — ICML 2026 regular_

### Official Review · Reviewer_CPAf · 2026-02-15

**Soundness:** 3
**Presentation:** 4
**Significance:** 3
**Originality:** 2
**Overall Recommendation:** 4
**Confidence:** 4

**Summary:**

The authors introduce the work “SusVibes” to address a benchmarking gap in examining how often models / SWE-agent systems write insecure code that is prone to vulnerabilities. Prior works either don’t target this setting explicitly (SWE-bench style works), or formulate at a scope that is either too small or unrealistic (code security benchmarks). The authors address this by putting together a SWE-bench style collection pipeline with key augmentations around looking for pull requests that specifically address a vulnerability (as defined against a list of CWEs) along with methods for generating + calibrating supplementary fields such as the issue description / problem statement. The benchmark reveals a stark disparity in solve rate versus secure code across all models and SWE-agent scaffolds, and techniques around modifying the prompt to request secure code more explicitly reveals surprising trends of correct solutions becoming incorrect due to overcompliance.

**Compliance With Llm Reviewing Policy:**

Affirmed.

**Final Justification:**

Will retain my original score. I think the project is quite neat, but I think the novelty of it is somewhat limited, as there have been several vulnerabilities-in-SWE-bench style projects that already exist. I think the framing the authors attempt to put forth, which distinguishes vibe coding and SWE-bench style evaluation:

> SWE-bench measures whether the requested functionality is achieved. SusVibes asks a different question: when a vibe coding agent implements a new feature to satisfy a natural language request, how likely is it to introduce a CWE?

Makes some sense to me, but it seems that in practice, the evaluation is still SWE-bench-unit-tests + an additional set of tests checking for security violations. So although I can understand this argument in concept, in practice it does not seem this work actually fundamentally introduces a new form of evaluation. Rather, it's still very much an augmentation of SWE-bench in terms of its collection schema.

That said, beyond the debate around novelty, I feel the paper was executed quite well and still answers an interesting question that is quite relevant with the proliferation of vibe coding.

**Key Questions For Authors:**

1. [Section 3.1; Generating solution…] Does generating a feature request using the masked implementation + C_{-1} risk underspecifying the solution? I can see how this prevents leakage, but would the resulting issue be “not fair” in that it doesn’t account for the vulnerability? Or is the expectation that the model should account for it even if not explicitly stated? (Is this taken care of by [Section 3.2; Adaptive Task Verification]?)
2. [Line 379] “The inspection reveals…” So SWE-agent also has this bug? But why would a scaffold having a bug mean that the LM would produce bugs that are also present in the scaffold? (E.g. if a scaffold has XSS vulnerability, would we expect the LM to write more XSS vulnerable code?)
3. What accounts for the differences in scaffold performance for funcpass? E.g. across the scaffolds, SWE-agent seems to have the highest solve rate - what accounts for this?
4. In appendix section B, I can see that it’s stated that different models / scaffolds account for different CWEs to differing degrees - why is this? What makes one model / scaffold more/less inclined to generating functional and/or secure code?

**Limitations:**

Yes

**Strengths And Weaknesses:**

**Strengths**

- [Section 1] The paper is well motivated, studying how often models are generating vulnerable code in a more realistic setting seems quite important.
- [General] The figures are nicely done - made a lot of the definitions and formulations easy to understand, and I personally like the aesthetic a lot as well!
- [Section 3] The task collection pipeline makes great use of existing works. The ability to automate such collection is a meaningful plus.
- [Section 4] The main results demonstrate that these concerns are meaningful, reflected by low performance on the benchmark.

[Overall] I enjoyed reading this work. I think the motivation, task formulation, and collection were all quite well defined, and the results in general confirm the authors’ motivation, providing a great testbed for examining these issues.

**Weaknesses**

- [Section 1;3] The definition of “vibe coding” could be grounded more concretely in prior research. I agree with the author’s definition, but it’d be worthwhile to map the terminology to prior research literature. To some audiences, I imagine “vibe coding” is much more of a colloquial term than a rigorous definition. The motivation can be more scoped as how existing NL2Code benchmarks either don’t examine security, or do so at a limited scope.
- [Section 3.3] Two of these upsides (Real-world SWE tasks, Scalability/Extensibility) seem to repeat SWE-bench. Most of the features are stated in distinction to code security benchmarks. Are we certain that existing SWE-bench style benchmarks don’t already capture some of the things here? E.g. SWE-bench has a large number of Django tasks. (More highlighting of why related works along this dimensions don’t capture things would help)
- [Section 4.2] These are interesting findings, but I’d like to get a sense of a “why” for a lot of these. E.g. “OpenHands is more secure than SWE-agent” - the SecPass is consistently higher across all models, based on Table 3, but why is this the case? From a more adversarial standpoint, what if this is simply a case of the task instances that OpenHands solved just being “easier” than SWE-agent wrt writing secure implementations?
- [Section 4.2; For tasks with the same…] I didn’t quite understand this. “Agents’s performance w/ the same CWE tag”, but Table 4 lists repository names? My understanding is that repository name != CWE tag, so how is Table 4 communicating this? When you say “projects with similar vulnerability types”, what does this mean / how is it quantified?

[Overall] I think the definition of vibe coding should be more rigorous, and the novelty propositions would be strengthened by more comparisons to SWE-* style work that this benchmark seems heavily inspired by. For Section 4.2, many of the results are interesting, but I would appreciate more explanations regarding why we are seeing such numbers. For instance, what would make one scaffold “more secure” than the other?

**Miscellaneous**

- [Line 101] “mi7igations” misspelling
- [Section 3; Harness Security Tests] Seems like this is exactly what SWE-bench does? Not that this needs to be stated, but is there anything different?
- [Table 3] What are the error bounds on this table / across how many runs were these averages calculated?
- [Section 4.1] Any particular reason why those 3 models?
- [Section 4.2; Gemini 2.5 Pro is the most…] How many task instances make up this intersection of “functionality correct subset of tasks”)
- [Table 5] How is this different from Table 4 / why not combine the two and add Gemini 3 Pro as a row?
- [Line 359] “analyze a code” -> “analyze code”
- [Figure 4] How is this matrix computed? Is this across all 200 predictions for a specific model/scaffold combo? Or across everything (in which case what is everything?)

Suggested Citations:
- [Related Work; Coding Agents] SWE-smith (https://arxiv.org/abs/2504.21798), BugPilot (https://arxiv.org/abs/2510.19898)
- [Related Work; Code Security] This is up to you, but I think prior work around using SWE-agent’s to identify / exploit security vulnerabilities (e.g., CyBench (https://arxiv.org/abs/2408.08926), InterCode-CTF (https://openreview.net/pdf?id=KOZwk7BFc3), CTF-Dojo (https://arxiv.org/abs/2508.18370)) could be worth briefly comparing against to distinguish.

---

> ### Author Rebuttal · Authors · 2026-03-31
>
> We thank the reviewer for the positive and detailed feedback. We are happy that the reviewer enjoyed reading this work and found the motivation, task formulation, and results well presented. We address the concerns below and will revise the manuscript accordingly.
>
> **W1: The definition of “vibe coding”**
>
> A1: **Definition of vibe coding:** It refers to using agents to produce software by describing goals in natural language with minimal review of the generated code, applicable to both professional developers and non-technical users[1,2].
>
> **Why existing NL2Code benchmarks fall short:** Three key gaps exist. First, benchmarks like HumanEval and MBPP focus on single-function implementation with limited context, whereas vibe coding requires full codebase inspection and multi-file edits. Second, NL2Code benchmarks provide concrete implementation descriptions, while vibe coding tasks are goal-oriented with no detailed implementation guidance. Third, NL2Code benchmarks generate code in a single round, whereas vibe coding agents interact iteratively with the executable environment to obtain feedback and refine their solutions.
>
> [1] Vibe Coding in Practice: Motivations, Challenges, and a Future Outlook--a Grey Literature Review
>
> [2] Vibe coding: programming through conversation with artificial intelligence
>
> **W2: [Section 3.3] & W5: Different between SusVibes and SWE-bench style benchmark**
>
> A2: **Security-oriented task construction:** Unlike SWE-bench style benchmarks that focus on functionality, SusVibes specifically targets security-relevant tasks by filtering commits starting from vulnerability-fix commits, ensuring that the selected functionalities are known to trigger real vulnerabilities. Tasks are further filtered to retain only those with human-written security tests, and executable environments are built to satisfy two conditions: before the fix, only functionality tests pass; after the fix, both functionality and security tests pass.
>
> **Vibe coding simulation:** SusVibes simulates real-world vibe coding scenarios where users specify only what functionality they want, without awareness of the underlying security risk. This is achieved by masking the original vulnerable implementation and adaptively revising the task description to cover full functionality requirements without leaking any security-related information.
>
> **W3: [Section 4.2] More insights on our findings & Q3: What accounts for the differences in scaffold performance?**
>
> A3: Please refer to our response to **Reviewer r8Dc’s W2.**
>
> **W4: [Section 4.2] Table 4. “Agents’s performance w/ the same CWE tag”**
>
> A4: Please refer to our response to **Reviewer d3j1‘s W6**.
>
> **W5: [Table 3] What are the error bounds on this table / across how many runs were these averages calculated?**
>
> A5: As stated on Line 271-274, we use pass@1 for FuncPass and SecPass because it can reflect real-world usage of vibe coding agents. Due to the long interaction (170+ turns on average) and the cost, the user usually cares more about the pass@1 instead of running it multiple times.
>
> **W6: [Section 4.1] Any particular reason why those 3 models?**
>
> W6: We choose Gemini 2.5 Pro, Claude 4 Sonnet and Kimi K2 because they are all reported with great agentic capabilities and cover both closed-source and open-sourced models. More LLMs’ results can be found in our response to **Reviewer** **d3j1’s W2.a.**
>
> **W7: [Section 4.2] How many task instances in this intersection of “functionality correct subset of tasks”**
>
> A7: For each pair of two models, we first identify the set of instances that both models get functionally correct, then compute each model's correct-and-secure ratio (SecPass ⟂ FuncPass) on that shared subset. The pairwise intersection sizes are: Gemini vs Claude = 34, Gemini vs Kimi = 32, and Claude vs Kimi = 65.
>
> **W8: [Table 5]  & Typos**
>
> A8: We split Table 1 and Table 5 because we would like to highlight within-family comparison. We will merge these two tables and also correct the other typos.
>
> **W9: [Figure 4] How is this matrix computed?**
>
> A9: As stated in Section 5 line 355-356, experiments in the mitigation section were conducted on SWE-agent + Claude. Yes, the matrix in Figure 4 is hence for a specific model and scaffold, across all 200 tasks.
>
> **Q1: Does generating a feature request using the masked implementation + C_{-1} risk underspecifying the solution?**
>
> A: Our adaptive task verification is designed to handle this: ensure the task description is sufficient and necessary for the functionality, but does not leak the security risk explicitly. We provide a more detailed demonstration of this process in Figure 6.
>
> **Q2: [Line 379] “The inspection reveals…”**
>
> A: We use “SWE-agent’s implementation” to refer to the implementation generated by SWE-agent for the task, not to the implementation of the SWE-agent scaffold itself.
>
> **Suggested Citations:**
>
> We appreciate the suggested related work and will compare them with our SusVibes in our revision.

---

> > ### Author Rebuttal · Reviewer_CPAf · 2026-04-05
> >
> > Thanks to the authors for the thorough rebuttal. I will address the rebuttal point by point:
> >
> > * W1 - addressed
> > * W2 - This introduces a bit of contradiction to me. The authors' response helped with understanding the difference between SusVibes and SWE-bench, but the framing of how SusVibes elucidates hidden pitfalls in vibe coding now feels a bit misfit. SWE-bench / SusVibes style task instances have an issue statement which explicitly asks for some problem to be resolved. The solution is then explicitly verified by tests. The framing of this problem therefore feels a bit awkward because SWE-bench does not really measure "vibe coding" performance. I think the framing of the paper might be better served if motivated from the standpoint of examining how likely it is that models introduce CWEs in an effort to resolve GitHub issues.
> > * W3 - addressed, putting a thorough analysis of this in the paper would greatly benefit the novelty and reproducibility of this paper's findings.
> > * W4 - addressed
> > * W5 - So agents were run on the benchmark just a single time? Pass@1 can still be computed across `n` runs. I think it'd be helpful to run a couple times to understand the variance and Pass@[1 - k] trends.
> > * W6 - addressed, but a minor point - the exclusion of Qwen, GPT models, Minimax from this list still feels unaddressed given the selection criteria. Was it mainly budget constraints?
> > * W7 - addressed
> > * W8 - addressed
> > * W9 - addressed
> > * Q1 - addressed
> > * Q2 - addressed
> >
> > I will maintain my original score.

---

> > > ### Author Response · Authors · 2026-04-07
> > >
> > > We appreciate the reviewer's detailed feedback and feel glad that most of the concerns have been addressed. We would like to further address the remaining concerns as follows:
> > >
> > > **W2 - This introduces a bit of contradiction to me.  I think the framing of the paper might be better served if motivated from the standpoint of examining how likely it is that models introduce CWEs in an effort to resolve GitHub issues.**
> > >
> > > A2: We thank the reviewer for this thoughtful comment. We would like to clarify why we believe the vibe coding framing remains appropriate for SusVibes.
> > >
> > > **The key distinction is the agent's objective, not the task format.** SWE-bench measures whether the requested functionality is achieved. SusVibes asks a different question: when a vibe coding agent implements a new feature to satisfy a natural language request, how likely is it to introduce a CWE? The task format is similar, but the evaluation target is fundamentally different — security risk during autonomous feature implementation, which is the central concern of vibe coding in practice.
> > >
> > > **The vibe coding framing captures a real and distinct risk.** A developer reviewing their own code may catch security issues before committing. A vibe coding agent that autonomously implements and submits a solution bypasses this review step. SusVibes is designed to measure this specific risk, which is why the vibe coding framing is more appropriate than simply characterizing SusVibes as a security-augmented SWE-bench.
> > >
> > > We will add a paragraph in the paper to make this distinction explicit and address potential confusion between the task format and the evaluation goal.
> > >
> > > ---
> > >
> > > **W5 - So agents were run on the benchmark just a single time? Pass@1 can still be computed across `n` runs. I think it'd be helpful to run a couple of times to understand the variance and Pass@[1 - k] trends.**
> > >
> > > A3: We thank the reviewer for the suggestions and will evaluate this setting in our next version.
> > >
> > > ---
> > >
> > > **W6 - addressed, but a minor point - the exclusion of Qwen, GPT models, Minimax from this list still feels unaddressed given the selection criteria. Was it mainly budget constraints?**
> > >
> > > A6: We have the results of Qwen3-Coder-Next in the [table](https://anonymous.4open.science/r/susvibes-7793/more_llm_performance.md). For GPT models and Minimax series, we have not evaluated them because of the cost and time constraints. We will keep evaluating more in our next version.
> > >
> > > | Model | Framework | FunPass | SecPass |
> > > | --- | --- | --- | --- |
> > > | Qwen3-Coder-Next | SWE-agent | 29.5% | 9.0% |
> > > | Qwen3-Coder-Next | OpenHands | 61.5% | 48.0% |

---

### Official Review · Reviewer_TZPu · 2026-03-11

**Soundness:** 3
**Presentation:** 3
**Significance:** 3
**Originality:** 3
**Overall Recommendation:** 4
**Confidence:** 4

**Summary:**

The vibe coding paradigm is widely getting adopted in software development flows. It usually involves generating, fixing large code sizes across classes and files. The paper studies security concerns of vibe coding by building a comprehensive automated benchmark and evaluating it against coding agents and LLMs. It further discusses a few strategies to overcome vulnerabilities in vibe-coding generated code and its limitations. Overall, authors bring attention to an important area of research with real-world implications and proposes a benchmark - SUSVIBES to enable further research in this area.

**Compliance With Llm Reviewing Policy:**

Affirmed.

**Final Justification:**

As software development shifts toward "vibe coding"—generating and fixing large swaths of code across multiple files based on high-level intent—the security of such generated code becomes a critical concern. This paper proposes SUSVIBES, a benchmark that uses Docker-based environments to evaluate how coding agents handle 77 different Common Weakness Enumerations (CWEs). It specifically focuses on whether agents inadvertently introduce or fail to fix vulnerabilities during feature implementation.

**Strengths -**
- High Practical Relevance: The paper identifies a significant gap in existing benchmarks, which often fail to account for the multi-file, large-scale code updates typical of modern AI-assisted development.
- Comprehensive Coverage: SUSVIBES supports 77 CWEs, offering a broader scope than many current security benchmarks.
- Sound Methodology: The use of Docker containers for environment setup and validation ensures that the evaluation is grounded in executable, real-world conditions.
- Open Research Contribution: The authors have committed to releasing the benchmark and code, which will be a valuable asset for the software security research community.

**Weaknesses and Rebuttal -**
- The authors clarified the task description and "masking" process, pointing to a more detailed verification pipeline involving a "verifier agent" that ensures all security-fix lines are represented in the task requirements.
- Vulnerability Detection Rigor - A primary critique was that the benchmark relies on human-written tests ($C_0$), which might miss new vulnerabilities introduced by the LLM. Although more thorough vulnerability detection may lead to further substantiate authors' findings the current paper lacks from require rigor and common practices.

**Decision** -
It addresses critical area of research with strong problem formulation, benchmark and evaluation. However, paper can be further improved with more rigor in areas of vulnerability detection and evaluation framework. I retain my original score (weak accept)

**Key Questions For Authors:**

1. Can you please expand on adaptive task verification process? The logic seems to be brittle based on description so please share results/metrics of this component if available.

2. What other tools, methods were evaluated if any for vulnerability verification beyond the curated test harness? Tools like [STELP](https://arxiv.org/pdf/2601.05467) can be used to further strengthen CWE detection (especially newly introduced CWEs).

3. Will this benchmark be publicly available?

**Limitations:**

Yes

**Strengths And Weaknesses:**

**Strength -**
1. The authors have successfully brought attention to vulnerabilities and security risks with widely adopting vibe coding paradigm. The paper clearly demonstrates limitations of existing benchmarks, agentic patterns and advises caution in adoption of vibe coding in security critical applications.
2. The paper has clear problem formulation, proposed solution (benchmark) and evaluation.
3. The paper also addresses a major problem of vibe coding suitable (large code updates spanning multiple files, wider CWE support, etc.) which may help other researchers to foster improvements. (provided it’s publicly available)
4. Benchmark curation framework is easy to follow with some exceptions (mentioned below)
5. Use of docker containers for environment setup and test validation is practical and can be used in real world applications.
6. Overall the paper is easy to read and understand.

**Weakness -**
1. Benchmark curation: Few stages, like, masking could be better explained for quick understanding. Fig 2 seems to be incomplete as it doesn’t show how task descriptions are created.
2. Benchmark curation pipeline: The evaluation of pipeline results and ablation is missing. Without accessing code or its results it’s difficult to evaluate efficacy of the SUSVIBE.
3. The paper is depending on a test harness compiled during curation to test whether vibe-coding-based code is secure or not. This is limited vulnerability detection as newly generated code may introduce some other vulnerabilities not identified with C_0 code tests. Commonly used vulnerability detection tools like [STELP](https://arxiv.org/pdf/2601.05467), [CodeShield](https://github.com/meta-llama/PurpleLlama/tree/main/CodeShield) could be used to truly detect SECPASS. A quick evaluation with such tools or mentioning as future work will be helpful.
4. Continuing on limited test hardness for vulnerability detection - Paper could be further improved with adversarial attacks, red teaming research.

Minor - type on line # 107 (mi7igations)

---

> ### Author Rebuttal · Authors · 2026-03-31
>
> We thank the reviewer for the valuable suggestions. We are pleased that the reviewer recognizes the significance of addressing security risks in the vibe coding paradigm and acknowledges the practical value of our execution environment setup. We take the concerns raised seriously and address them below.
>
> **W1: Benchmark curation: masking could be better explained. Fig 2 is incomplete.**
>
> A1: Thanks for the suggestion. Figure 2 focuses on the general pipeline and leaves the detailed task description creation process to Figure 6. We will modify the figure and caption to make this point clearer.
>
> **W2: Benchmark curation pipeline: Without accessing code or its results it’s difficult to evaluate efficacy of the SUSVIBE. & Q3: Will this benchmark be publicly available?**
>
> A2: We have uploaded our SusVibes benchmark as well as our code to https://anonymous.4open.science/r/susvibes-submit-FE7C and will release it. We also conduct human evaluation on the quality of our generated tasks (see Q1).
>
> **W3: This is limited vulnerability detection as newly generated code may introduce some other vulnerabilities not identified with C_0 code tests.**
>
> A3: We agree that human-written security tests cannot guarantee full coverage of all possible vulnerabilities. We address this concern through the following efforts.
>
> - **Empirical validation via static analysis:** We apply CodeQL 2.23.5 with the python-security-quality.qls query, following the static analysis approach used in prior work (Asleep). We choose a representative subset of 20 tasks covering 12 CWEs across 18 real-world repositories. Across all tasks, we find that the agent did not introduce critical new vulnerabilities beyond the coverage of human-written tests.
> - **The limitation strengthens our main finding**: If coding agents do introduce additional vulnerabilities not covered by human-written tests, this would only make agent security performance appear worse than currently reported, reinforcing rather than undermining our central claim that current coding agents perform poorly on security.
> - **Broader coverage than existing benchmarks**: SusVibes covers 77 CWEs, outperforming comparable benchmarks, indicating that our overall coverage is already comprehensive in the current state of the field.
>
> We recognize that full vulnerability coverage remains an open challenge and are actively expanding the benchmark with additional human-annotated security test cases. We consider this an important direction for future work.
>
> **W4: Paper could be further improved with adversarial attacks, red teaming research.**
>
> W4: The vulnerability detection focuses on the existing vulnerable code, and adversarial attack and red-teaming focuses on exploiting the existing vulnerability. However, our SusVibes is to evaluate whether the vibe coding agent will introduce new vulnerability into during their new feature implementation, instead of detecting the existing ones in the codebase. We will discuss the difference and relationship in our revision.
>
> **Q1: adaptive task verification process?**
>
> A1: We put a concrete demonstration of the adaptive task verification process in *Figure 6* and the prompt is described as *Prompt: Verify Task Description with Secure Implementation of the Features* in Appendix page 13-14.
>
> Specifically, a verifier agent is used to check whether the generated task description covers all lines of feature implementation plus the security fixes, by linking each code line to a requirement in the description. If any line in the secure implementation is not mentioned by the task description, it will go back to regenerate a larger mask; otherwise, the task description and the mask will be returned.
>
> We also conduct human evaluation on a randomly sampled subset of 15 tasks (7%) to manually verify the quality. These 15 tasks span across 12 different GitHub projects, accessing 14 different CWEs, forming a representative subset of SusVibes.
>
> 3 software engineer annotators assess the quality of the mask generated by the LLM based on the following criteria:
>
> 1. **Valid Mask:** the mask purely deletes an implementation (or adds at most some place holders)
> 2. **Sufficient Mask:** The mask removes a sufficient feature implementation surrounding the security fix, containing enough security context.
>    1. If covers all lines that were touched by the security fix patch.
>    2. If covers the implementation of a feature that requires the security fix.
> 3. **Necessary Mask:** The mask avoids obviously unrelated or excessive deletions.
>
> The resulting table is shown in this [link](https://anonymous.4open.science/r/susvibes-7793/annotations.md). It is suggested by human software engineers that the masks constructed from the generation-verification pipeline in SusVibes are reasonable in forming security-oriented coding tasks.

---

> > ### Author Rebuttal · Reviewer_TZPu · 2026-04-03
> >
> > Thank you for your response. My questions are answered.

---

### Official Review · Reviewer_d3j1 · 2026-03-11

**Soundness:** 3
**Presentation:** 2
**Significance:** 3
**Originality:** 2
**Overall Recommendation:** 5
**Confidence:** 5

**Summary:**

The authors present SusBench, a variant of SWE-Bench that measures performance at adding features to real world repositories. The features are designed such that their original human implementation is insecure (this being verified through past vulnerability-patching commits) and accompanied by tests that verify correctness and security. The authors evaluate a 4 coding models across 3 coding agents ( among them very recent coding models) and observe that they achieve performance of less than 60% and security of less than 10%.

**Compliance With Llm Reviewing Policy:**

Affirmed.

**Final Justification:**

The authors provided extensive manual verification of their results and intend to improve the final benchmark significantly and an extensive evaluation on SOTA LLMs. This addresses my key concern with the paper overall.

**Key Questions For Authors:**

- Can you provide any details about manual investigations performed, or perform a manual investigation of the quality of the benchmark?

**Limitations:**

The authors acknowledge some limitations in the appendix. I would prefer this section to be in the main text. Some missed limitations:
- There is no human validation of the test cases, so the strength of the claim of low security is low
- The evaluation was performed only on a few models and does not allow systematic insights.

**Strengths And Weaknesses:**

Strengths

- *Presentation*: The figures are very clean.
- *Significance*: The paper tackles a highly relevant topic, namely the security of coding agents. They meaningfully advance the field by providing a broader scoped and more extensive benchmark than prior works (more instances, more and more varied CWEs, larger required edits). The focus on feature addition can also be highlighted since feature addition is rare for SWE-Bench and its variants.

Weaknesses

- *Soundness*: The authors describe that they source the security tests from human-written securit tests added at the time of the vulnerability patch. This leaves me wondering whether those security tests are general enough to accurately capture a) an alternative implementation that is secure but uses different logic to the human solution and b) captures other vulnerabilities- In BaxBench this is solved elegantly by testing end-to-end exploits (all human written). In SecureAgentBench the vulnerability can always lead to a crash, and they try to trigger this crash + they additionally use static analysis to check for new vulnerabilities. There seems no extra quality assurance conducted for SusBench. Can you provide a manual analysis of at least a representative sample of the security tests to confirm that they meaningfully only report true vulnerablities?
- *Soundness*: The evaluation is very thin: only 4 models are executed, among them only 1 open weight model. 2 models of the same family (gemini 3, gemini 2.5). Since the only other contribution of the paper is the benchmark, I would expect an extensive overview over the field of current LLM families (including OpenAI, DeepSeek, Qwen, ...) and an analysis of a few differently sized models of the same family (e.g. Qwen 3 8B, 14B, 32B) to derive trends or other insights. BaxBench also includes one evaluation without any security reminder. I think this is relevant context to learn about the "standard" security of models, since most vibe coders will not even think about security. Always appending the security reminder can also skew the results for correctness, which might be much higher than reported here when the reminder is not present.
- *Presentation*: The paper presents itself as the only security and functionality benchmark that evaluates agent performance (line 28-37). This presentation is wrong in my opinion: All prior benchmarks *can* be solved using agents as well. BaxBench even presents results of coding agents, but shows that they fare similar to direct generation by LLMs. The execution environment is also not unique to SusBench - in fact every benchmark that executes its functions (including BaxBench and SecureAgentBench) necessarily provides the execution environment in which the functions/tests can be executed. As the naming suggests, SecureAgentBench as a matter of fact actually also evaluates coding agents. Finally the agent framing mismatches with the experimental results: There is also no agent specific analysis in the experimental results - no analysis of tool calls, traces, etc. In my opinion a better presentation is that the benchmark is more challenging by requiring larger edits, larger context handling. This in itself is a strong contribution (as mentioned in the strenghts).
- *Presentation*: In the same vein the authors should not claim that prior work is single-file (line 25) and then contradict themselves in Table 1
*Originality* is not a strength of this paper. Most of the benchmark construction and design is very similar to the endless varieties of SWE-Bench. But I think this can be glossed over for a well executed benchmark that provides a large amount of high-quality, difficult tasks. Whether the presented work is such a benchmark is unclear however due to the missing human validation and quality assurance. The lack of experimental breadth makes it difficult to obtain relevant insights.
- *Significance*: The "Qualitative Analysis" in section 4.3 is just a concrete example of an issue. I think it should include the test case and how the test case catches the insecure implementation or functional errors if there are any. A good qualitative analysis should aggregate a number of insights by inspecting manually several examples, or highlight a concrete case in detail. For example, is there a clear difference between the model behavior on some task with self-select/oracle/generic reminders? How does this express? What is the effect on functionality/security?
- *Presentation*: Table 4 is very hard to understand. What are each values, what are we supposed to learn from the table? Why is 100.0 bolded for claude 4 sonnet but not gemini 2.5 pro which is also 100 in py-libn? I also don't understand what the paragraph in lines 318-324 mean, in particular what does it mean that Claude can not ensure a more secure implementation than Gemini?
- *Presentation*: Table 1 should maybe show whether the model output is executed / checked for correctness or just security (e.g. Pearce only evaluate Security)
- *Presentation*: Table 5 should be merged with Table 3. Also the current layout of Table 5 makes it hard to compare values that should be compared because they are not next to each other. The same is currently true for comparing the impact of agent harnesses in Table 3
- *Nitpick*: There is a 7 in mi7igations in line 101
- *Nitpick*: Lines 326-329 use once Gemini 3.0 and once Gemini 3 - this should be consistent throughout the paper.


The authors would IMO do themselves a service in condensing the existing content and adding more insights or a discussion section to the paper.

---

> ### Author Rebuttal · Authors · 2026-03-31
>
> We thank the reviewer for the valuable and thoughtful evaluation, which can help us improve the work. We are glad that the reviewer recognizes the contributions of this work. We appreciate the concerns, discuss them as follows, and will revise the manuscript accordingly.
>
> **W1.a. *Soundness*: Can you provide a manual analysis of at least a representative sample of the security tests to confirm that they meaningfully only report true vulnerabilities?**
>
> A1.a: We provide manual analysis of a set of two SusVibes’ security tests, covering two different CWEs in this [link](https://anonymous.4open.science/r/susvibes-7793/test_analysis.md).
>
> **W1.b: captures other vulnerabilities.**
>
> A1.b: We kindly refer to the reviewer to the response to **W3 of Reviewer TZPu**.
>
> **W2.a.: More LLMs.**
>
> A2.a: We further evaluate more LLM families such as GLM-4.7-Flash (30B MoE), GLM-5 (744B MoE) and Qwen3-Coder-Next (80B MoE). We can find that within the same model family (GLM), the larger model shows better performance in both FuncPass and SecPass. Find the results in this [link](https://anonymous.4open.science/r/susvibes-7793/more_llm_performance.md).
>
> **W2.b.: Evaluation with no security reminder.**
>
> A2.b:  This current generic prompt is “Make sure to follow best security practices and avoid common vulnerabilities” (in Appendix line 936-940)*,* which is general enough to avoid any detailed security information. We will also complement this no-reminder most insecure setting in the revision.
>
> **W3: The paper presents itself as the only security and functionality benchmark that evaluates agent performance (line 28-37).**
>
> A3: We thank the reviewer for the valuable suggestions on the presentation issues in our paper. We didn’t claim that our SusVibes is the first and only to benchmark agent performance on both functionality and security, nor the only one that provides executable environment. In our manuscript (Line 28-37), we described some limitations of previous code security benchmarks. We will make our wording more rigorous.
>
> As we stated in our contributions (Line 71-86), our SusVibes provides a larger CWE coverage and more required multi-file edits in 200 tasks, and evaluates different combinations among agentic LLMs and agent framework to investigate the vibe coding security issue.
>
> We'd like to highlight the differences with prior works in the table in the following [link](https://anonymous.4open.science/r/susvibes-7793/table1.md).
>
> **W4: should not claim that prior work is single-file (line 25) & Q1: manual investigation of the quality**
>
> A4: We thank the reviewer for raising this point. We will revise this statement to “most of their contexts are limited to a single file or function” to be more accurate.
>
> We refer the reviewer to **our response to Q1 of Reviewer TZPu**.
>
> **W5: is there a clear difference between the model behavior on some task with self-select/oracle/generic reminders?**
>
> A5: We performed a preliminary analysis of the trajectories under these strategies, which suggests several patterns as follows.
>
> **Cognitive Overload and Over-constraint (Generic vs. Self-Selection)**
>
> Forcing the agent to read a large CWE list, assess relevance, and justify a selection before implementing code splits its attention and degrades performance. The added burden also causes the agent to over-constrain its solution out of excessive caution, leading to a sharp increase in functional failures.
>
> **Explicit Cues Reduce Search Entropy (Oracle vs. Self-selection)**
>
> The Oracle strategy directly specifies the target vulnerability, allowing the agent to skip the broad conceptual search and focus immediately on how to secure the implementation. This targeted focus improves secure hit rates compared to self-selection.
>
> **W6: Table 4 is very hard to understand.**
>
> A6: We thank the reviewer for raising this point and will ensure its clearness in our revision.
>
> - **CWE type matters but is not the only factor:** While Appendix results show that CWE type influences how likely agents are to generate vulnerable code, Table 4 investigates whether project context is an additional contributing factor.
> - **Table 4 controls for CWE type to isolate project context:** By selecting tasks with similar vulnerability types and grouping them by project, the analysis finds that FuncPass and SecPass behave differently across projects. Claude consistently outperforms Gemini on FuncPass across projects, but SecPass shows no consistent winner, indicating that project context meaningfully affects security performance.
> - **Bolding correction:** The original bolding was intended to highlight that Claude is consistently "not worse" than Gemini on FuncPass.
>
> We’ll make sure that the preliminary finding is well-presented, and the bolding issue is corrected in our revision.
>
> **W7:  Table 1**
>
> A7: We add a clearer comparison in the table in W3.
>
> **W8: *Nitpick* & Typo**
>
> A8: Thanks for pointing out. We will revise the table and typos in our manuscript.

---

> > ### Author Rebuttal · Reviewer_d3j1 · 2026-04-03
> >
> > I thank the authors for their clarifications.
> >
> > W1: I thank the authors for the provided examples, they indeed look promising. I would still strongly prefer a more extensive human verification on more than 2 samples (say, around 10-20% of the dataset), just to verify that the tests are general. Since the test cases are the main contribution of the paper, I think they should be adequately checked. I may have missed that such a verification was conducted in response to any of the other reviews.
> >
> > W3: I was referring to two paragraphs in line 31. I suggest them to be weakened - for example BaxBench does evaluate agents in multiple turns and in execution environments.

---

> > > ### Author Response · Authors · 2026-04-07
> > >
> > > **W1: I thank the authors for the provided examples; they indeed look promising. I would still strongly prefer a more extensive human verification on more than 2 samples (say, around 10-20% of the dataset), just to verify that the tests are general. Since the test cases are the main contribution of the paper, I think they should be adequately checked. I may have missed that such a verification was conducted in response to any of the other reviews.**
> > >
> > > A: We sincerely thank the reviewer for this feedback and have conducted a substantially expanded human verification study to address the concern about sample scale.
> > >
> > > We recruited 5 software engineering experts to independently evaluate the entire 200 tasks using the following rubric:
> > >
> > > - **Pass (fully covers):** The test verifies the security property itself, independent of implementation details
> > > - **Conditional Pass (partially covers):** The test checks the correct security property, but binds to incidental implementation details
> > > - **Fail (cannot cover):** The test is coupled to the golden implementation rather than the security property
> > >
> > > For Conditional Pass and Fail cases, annotators additionally judged whether the test could be straightforwardly rewritten to achieve full coverage (*Easy to Refine: Yes/No*).
> > >
> > > Finally, we find that 6.5% of the security test cases are coupled to the golden human-written implementation, which may affect their verification capabilities of other security implementations. **The remaining 93.5% all provide verification for the security properties.**
> > >
> > > Among them, 49.5% fully cover all potential security implementations. The remaining 44.0% may miss some edge cases, but are fixable — 92.1% of them can achieve full coverage with small edits.
> > >
> > > Based on the verification, we use the 49.5% tasks with full coverage security test cases as the SusVibes-verified subset and report the evaluation results below. The experimental results show the same trend as what we report in our paper.
> > >
> > > **FuncPass**
> > >
> > > | Model | SWE-agent | OpenHands | Claude Code |
> > > | --- | --- | --- | --- |
> > > | Claude 4 Sonnet | 67.7 | 52.5 | 52.5 |
> > > | Kimi K2 | 26.3 | 42.4 | 46.5 |
> > > | Gemini 2.5 Pro | 28.3 | 29.3 | 20.2 |
> > >
> > > **SecPass**
> > >
> > > | Model | SWE-agent | OpenHands | Claude Code |
> > > | --- | --- | --- | --- |
> > > | Claude 4 Sonnet | 13.1 | 14.1 | 7.1 |
> > > | Kimi K2 | 7.1 | 9.1 | 8.1 |
> > > | Gemini 2.5 Pro | 9.1 | 11.1 | 5.1 |
> > >
> > >
> > > We will improve the coverage of the 44.0% tasks with more test cases, and update the failed tasks in our next version.
> > >
> > > ---
> > >
> > > **W3: I was referring to two paragraphs in line 31. I suggest them to be weakened - for example BaxBench does evaluate agents in multiple turns and in execution environments.**
> > >
> > > A3: We thank the reviewer for this explanation. Our intended point was: that existing benchmarks, taken as a landscape, leave the specific intersection of vibe coding, CWE introduction, and multi-file repository-level edits underexplored.
> > >
> > > We will reframe lines 28–37 to reflect this more modest and accurate characterization, clarifying that SusVibes differentiates itself through its focus on CWE coverage breadth, multi-file editing requirements, and systematic evaluation of vibe coding agent combinations — rather than through properties BaxBench already provides.

---

### Official Review · Reviewer_r8Dc · 2026-03-12

**Soundness:** 1
**Presentation:** 3
**Significance:** 2
**Originality:** 2
**Overall Recommendation:** 3
**Confidence:** 4

**Summary:**

This paper introduces SuSVIBEs, a new benchmark for evaluating code agents on their ability to generate secure code. The benchmark is constructed through an automatic pipeline that mines real-world vulnerability‑fix commits from open‑source repositories, reverts to the vulnerable version, harnesses both functionality and security tests from the commit history, and then uses agents themselves to mask out the feature implementation and generate natural language task descriptions. Compared to existing benchmarks, tasks in SuSVIBEs are more complex, operating at the repository level with multi‑file edits, covering a broader range of vulnerability types (77 CWEs), and assessing multi‑turn interactions of vibe‑coding agents. The evaluation results show that frontier LLMs and software engineering agents perform poorly on secure code generation, with over 80% of functionally correct solutions failing security tests.

**Compliance With Llm Reviewing Policy:**

Affirmed.

**Final Justification:**

Thanks the author for their feedback. Specifically, they provide the distribution of cross-file tasks to erase my first main concern. However, for my first major concern, I feel less convinced due to the small scale of the samples (only 7%), even less than 10%. Actually, since the core contribution of the paper is the proposed dataset, I think their qualities should be more rigorously guaranteed. I also see other reviewers share similar concerns.

Therefore, by considering the submission as well as the rebuttal overall, I would remain my original score, and lean for rejection (score 3: weak rejection).

**Key Questions For Authors:**

1. Did the authors manually inspect the agent-constructed tasks in SUSVIBES?
2. How many cross-file tasks in SUSVIBES?

**Limitations:**

Not fully

**Strengths And Weaknesses:**

**Strengths**

- Significance:

The problem addressed in this paper is important and practically meaningful. The paper tackles a problem of immense and growing practical importance. As "vibe coding" and AI agents become more integrated into software development workflows, the potential for introducing security vulnerabilities at scale is a major concern. And comparing with existing benchmarks, it covers a significantly larger number of CWE types (77) and requires more complex, multi-file edits.
- Presentation:
This paper is generally well-structured and easy to follow.

**Weaknesses**

- Soundness:

I am concerned about the reliability of the benchmark construction pipeline, which relies heavily on LLM agents (SWE-Agent) at multiple steps: mask generation, task description creation, and verification. While using agents to build benchmarks is innovative, their inherent unreliability raises questions about the resulting dataset quality.

  The most telling issue emerges when I am examining the motivating example in detail (Figure 2 and Figure 8). The original vulnerability fix commit C₀ addresses a timing side-channel attack in Django's authentication component by introducing a `fake_runtime` mechanism that equalizes response times between successful and failed login attempts. This is the core security fix and the very reason this task was selected for the benchmark.

  However, the generated task description in Figure 8 completely omits this security requirement. The description only mentions functional aspects: implementing `verify_password()` to return a tuple of booleans, handling edge cases like `None` passwords, and supporting algorithm changes. There is no mention of constant-time execution, timing attack prevention, or the need to avoid early returns, which is precisely the vulnerability that agents later reintroduce. This reveals a fundamental problem: if the task description does not communicate the security requirement, then agents failing to implement security measures cannot be fairly judged as "insecure". They are simply following the instructions they were given. The paper later shows agents introducing the timing vulnerability, but this is exactly what the description implicitly allows, even encourages, by only specifying functional behavior.

For short, it remains unclear whether SuSVIBEs measures agent security capability or simply reflects the quality of automatically generated task descriptions.

- Evaluation:

First, the paper observes that OpenHands produces more secure code than SWE-Agent when using the same underlying LLM. However, it stops at reporting this finding without exploring why this difference exists. A deeper investigation into the agent architectures could provide valuable insights for designing more secure agents.

Second, the paper emphasizes that SUSVIBES is more complex than existing benchmarks because it involves repository-level, cross-file tasks. However, all the detailed examples provided in the qualitative analysis (Django, Buildbot, Wagtail, aiohttp-session) are confined to single files and single functions. To substantiate the complexity claim, the authors should provide a clear distribution of single-file vs. multi-file tasks in the benchmark and analyze whether agent performance differs significantly between these two categories. It will be better if the authors include at least one detailed case study of a genuinely cross-file task to illustrate the additional challenges.

---

> ### Author Rebuttal · Authors · 2026-03-31
>
> We thank the reviewer for the valuable suggestions. We are happy to see that the reviewer thinks our vibe coding problem is critical and timely. We discuss them as follows, and will address them further in our manuscript.
>
> **W1: task description in Figure 8 completely omits this security requirement.**
>
> A1: The security requirement is omitted in the task description intentionally in order to simulate the real-world scenario for vibe coding:
>
> (i) Vibe coding users can be non-software developers without coding knowledge[1,2]. In Figure 8, it is more common for the users to let the vibe coding agent implement the verify_password() function without knowing that it may cause timing side channel risk.
>
> (ii) Vibe coding agents should have the capability to provide the secure feature implementation without the explicit secure clues. This is similar to the secure function-level code generation. For example, when implementing C/C++ functions, the code solution should never trigger security risks such as overflow without explicit hints. This requires the agent to automatically be aware of the potential security risks involved.
>
> (iii) We evaluate agents given the task description without explicitly hinting at the security risks. However, we also experimented adding the security concerns to the task description in our mitigation experiments in Section 5. We note that this setting aligns more closely with the reviewer’s suggested eval scenario, where security requirements are explicitly provided in the prompt. However, this is not the primary purpose of our benchmark.
>
> As a result, we find that adding these additional security cues is not always helpful for agents to produce correct-and-secure code.
>
> [1] Vibe Coding in Practice: Motivations, Challenges, and a Future Outlook – a Grey Literature Review
>
> [2] Position: Humans are Missing from AI Coding Agent Research
>
> ---
>
> **W2: It stops at reporting this finding without exploring why this difference exists.**
>
> A2: We appreciate this insightful suggestion. We agree that the performance gap between OpenHands and SWE-agent under the same backbone LLM is an important finding. While a full causal disentanglement is beyond the scope of this paper, we performed a detailed trajectory analysis of SWE-agent and OpenHands on a subset of tasks, which suggests several plausible hypotheses for this gap.
>
> - **Meta-cognition:** OpenHands' explicit reasoning tools (`think`, `task_tracking`) encourage more deliberate planning before editing. OpenHands more often pauses to plan between actions and follows slightly longer trajectories on average (about 178 steps vs. 166 for SWE-agent). By contract, SWE-agent's tighter loop biases toward immediate local fixes that may miss security  considerations, and long term planning for security checks.
> - **Context exposure:** SWE-agent's constrained ACI (`str_replace_editor`, `grep`, `find`) encourages inspecting, and editing code in controlled, small increments (localized search-and-edit behavior). OpenHands uses a more general design, where the model interacts through flexible shell commands. It surfaces broader code context, which helps when vulnerabilities span multiple components.
> - **Tool-task fit:** SWE-agent works better for dispersed syntactic fixes requiring repeated search, while OpenHands excels when vulnerabilities require centralized contextual reasoning. For example, `grep` accounts for a large portion of SWE-agent's tool calls, making it effective when a secure fix requires locating and consistently updating many dispersed sites, whereas OpenHands' flexible bash-based interaction is better suited for vulnerabilities tied to centralized logic components.
>
> **W3: single-file vs. multi-file tasks & Q2: How many cross-file tasks ?**
>
> A3: We will report the distribution of the file count of all tasks, how it relates to the performance in Section 4.2 in our manuscript, and a qualitative example in Appendix D.
>
> We were only showing detailed single-file examples for easy visualization. The tasks in SusVibes require editing 1.8 files and 170 lines on average as shown in Table 2, 32% of the tasks are multi-file, and 18% of the tasks require more than two files.
>
> We further break down agents’ performance on tasks requiring editing different numbers of files. There is a clear tendency that as the number of files increases, both the functional and the security performance of agents decreases, indicating the gradually-increasing difficulty level and challenges. Find the results in this [link](https://anonymous.4open.science/r/susvibes-7793/multi_file_performance.md).
>
> **Q1: Did the authors manually inspect the agent-constructed tasks in SusVibes?**
>
> A1: We conduct human evaluation on a randomly sampled subset of 15 tasks (7%) to manually verify the quality. We kindly refer the reviewer to our response to **the Q1 of Reviewer TZPu**.

---

> > ### Author Rebuttal · Reviewer_r8Dc · 2026-04-04
> >
> > Thanks the author for their feedback.
> >
> > Specifically, they provide the distribution of cross-file tasks to erase my first main concern. However, for my first major concern, I feel less convinced due to the small scale of the samples (only 7%), even less than 10%. Actually, since the core contribution of the paper is the proposed dataset, I think their qualities should be more rigorously guaranteed.  I also see other reviewers share similar concerns. Therefore, I would remain my original score.

---

> > > ### Author Response · Authors · 2026-04-07
> > >
> > > We sincerely thank the reviewer for this feedback and have conducted a substantially expanded human verification study to address the concern about sample scale.
> > >
> > > We recruited 5 software engineering experts to independently evaluate the entire 200 tasks using the following rubric:
> > >
> > > - **Pass (fully covers):** The test verifies the security property itself, independent of implementation details
> > > - **Conditional Pass (partially covers):** The test checks the correct security property, but binds to incidental implementation details
> > > - **Fail (cannot cover):** The test is coupled to the golden implementation rather than the security property
> > >
> > > For Conditional Pass and Fail cases, annotators additionally judged whether the test could be straightforwardly rewritten to achieve full coverage (*Easy to Refine: Yes/No*).
> > >
> > > Finally, we find that 6.5% of the security test cases are coupled to the golden human-written implementation, which may affect their verification capabilities of other security implementations. **The remaining 93.5% all provide verification for the security properties.**
> > >
> > > Among them, 49.5% fully cover all potential security implementations. The remaining 44.0% may miss some edge cases, but are fixable — 92.1% of them can achieve full coverage with small edits.
> > >
> > > Based on the verification, we use the 49.5% tasks with full coverage security test cases as the SusVibes-verified subset and report the evaluation results below. The experimental results show the same trend as what we report in our paper.
> > >
> > > **FuncPass**
> > >
> > > | Model | SWE-agent | OpenHands | Claude Code |
> > > | --- | --- | --- | --- |
> > > | Claude 4 Sonnet | 67.7 | 52.5 | 52.5 |
> > > | Kimi K2 | 26.3 | 42.4 | 46.5 |
> > > | Gemini 2.5 Pro | 28.3 | 29.3 | 20.2 |
> > >
> > > **SecPass**
> > >
> > > | Model | SWE-agent | OpenHands | Claude Code |
> > > | --- | --- | --- | --- |
> > > | Claude 4 Sonnet | 13.1 | 14.1 | 7.1 |
> > > | Kimi K2 | 7.1 | 9.1 | 8.1 |
> > > | Gemini 2.5 Pro | 9.1 | 11.1 | 5.1 |
> > >
> > > ---
> > >
> > > We will improve the coverage of the 44.0% tasks with more test cases, and update the failed tasks in our next version.

---

### Decision · Program_Chairs · 2026-04-30

**Decision:**

Accept (regular)

**Comment:**

This paper introduces SuSVIBEs, a benchmark for evaluating the security of code generation agents in realistic, repository-level settings. By mining real-world vulnerability-fix commits and constructing executable tasks, it targets an important and timely problem.

The reviewers provided mixed but overall positive assessments (two Weak Accepts, one Accept, one Weak Reject). All reviewers agreed on the significance of studying security in “vibe coding” and appreciated the benchmark’s scale, CWE coverage, and practical setup. The main concerns centered on the reliability of the automatically constructed tasks and the adequacy of human validation.

During the rebuttal, the authors substantially strengthened their evidence by conducting a full-scale human verification study over all tasks, showing that most test cases meaningfully capture security properties and releasing a verified high-quality subset with consistent results. Additional clarifications on task design, multi-file complexity, and evaluation further addressed reviewer questions.

While some reviewers maintained minor reservations about validation rigor and evaluation breadth, the majority concerns were resolved. Given the importance of the problem and the value of the benchmark to the community, I recommend Accept.